# Naturally acquired antibodies against 4 *Streptococcus pneumoniae* serotypes in Pakistani adults with type 2 diabetes mellitus

Izaz Ahmad[1], Robert Burton[2], Moon Nahm[2], Hafiz Gohar Ejaz[1], Rozina Arshad[3], Bilal Bin Younis[3], Shaper Mirza[1]*

1 Department of Life Sciences, Syed Babar Ali School of Science and Engineering, Lahore University of Management Sciences, Lahore, Pakistan, 2 Division of Pulmonary, Allergy, and Critical Care Medicine, University of Alabama at Birmingham, Birmingham, AL, United States of America, 3 Sakina Institute of Diabetes and Endocrine Research, Shalamar Hospital Lahore, Lahore, Pakistan

* shaper.mirza@lums.edu.pk

**Data Availability Statement:** All relevant data are within the manuscript. Also, additional data are provided in supplementary docs.

## Abstract

Immune response elicited during pneumococcal carriage has been shown to protect against subsequent colonization and infection by *Streptococcus pneumoniae*. The study was designed to measure the baseline serotype-specific anti-capsular IgG concentration and opsonic titers elicited in response to asymptomatic carriage in adults with and without type 2-diabetes. Level of IgG to capsular polysaccharide was measured in a total of 176 samples (124 with type 2 diabetes and 52 without type 2 diabetes) against serotype 1, 19F, 9V, and 18C. From within 176 samples, a nested cohort of 39 samples was selected for measuring the functional capacity of antibodies by measuring opsonic titer to serotypes 19F, 9V, and 18C. Next, we measured levels of IgG to PspA in 90 samples from individuals with and without diabetes (22 non-diabetes and 68 diabetes). Our results demonstrated comparable IgG titers against all serotypes between those with and without type 2-diabetes. Overall, we observed higher opsonic titers in those without diabetes as compared to individuals with diabetes for serotypes 19F and 9V. The opsonic titers for 19F and 9V significantly negatively correlated with HbA1c. For 19F, 41.66% (n = 10) showed opsonic titers ≥ 1:8 in the diabetes group as compared to 66.66% (n = 10) in the non-diabetes group. The percentage was 29.6% (n = 7) vs 66.66% (n = 10) for 9V and 70.83% (n = 17) vs 80% (n = 12) for 18C in diabetes and non-diabetes groups respectively. A comparable anti-PspA IgG (p = 0.409) was observed in those with and without diabetes, indicating that response to protein antigen is likely to remain intact in those with diabetes. In conclusion, we demonstrated comparable IgG titers to both capsular polysaccharide and protein antigens in those with and without diabetes, however, the protective capacity of antibodies differed between the two groups.

## Introduction

The incidence of diabetes is increasing at an exponential rate globally. Data from the International Diabetes Federation (IDF) demonstrate that approximately 537 million people (20–79

**Funding:** This study was supported by National Research Program for Universities (NRPU) Higher Education Commission (HEC)-5931- and Faculty Incentive Fund (FIF), Lahore University of Management Sciences. The funding was awarded to the corresponding author, SM. The funders had no role in study design, data collection and analysis, decision to publish, or preparation of the manuscript.

**Competing interests:** The authors have declared that no competing interests exist.

years) are currently living with diabetes (over 90% of people with type 2 diabetes), and the number is projected to rise to 643 million by 2030. With a total of 33 million adults living with diabetes, Pakistan is ranked 3rd in the world, and the number is projected to rise to 62.2 million by 2045 [1].

Type 2-diabetes is a risk factor for several upper and lower respiratory tract infections, particularly community-acquired pneumonia [2]. *Streptococcus pneumoniae* (pneumococcus), an encapsulated, extracellular bacterium, has been implicated in at least 22% of community-acquired pneumonia globally. Asymptomatic colonization of nasopharyngeal mucosa is a prerequisite for all infections caused by *S. pneumoniae*. Pneumococcus evades host immune defenses and can cause both invasive (septicemia, meningitis, pericarditis) and non-invasive (sinusitis, otitis media, pneumonia) infections [3, 4]. Invasive pneumococcal diseases (IPD) mainly affect young children, older adults, and individuals with co-morbidities like cardiovascular diseases and diabetes [5–8]. Studies show that individuals with diabetes have a 3.5 times higher risk of IPD and a higher mortality rate as compared to those without diabetes [8–11]. This higher susceptibility of individuals with diabetes to pneumococcal infections is attributed to immune fragility reported in type-2 diabetes [12–20].

The protection against carriage and pneumococcal infection is multifactorial and has been shown to involve both antibody-mediated and cellular immune responses [21]. The asymptomatic colonization of pneumococcus or infection elicits antibody responses to capsular polysaccharides and protein antigens [22–25]. Pneumococcal capsule is an important virulence factor in the pathogenesis of pneumococcal disease as it facilitates the colonization of mucosal surfaces, inhibits complement pathways, and reduces opsonization [3, 26]. On the basis of capsular polysaccharides, *S. pneumoniae* can be divided into 100 different serotypes in that each serotype differs in the chemical composition of the capsule [27]. The concentration of antibodies to capsular polysaccharides and the functionality of antibodies against different serotypes included in the vaccine is used as a correlate of protection to establish the efficacy of pneumococcal vaccines among vaccinees [28–33]. While there are no established serological correlates of protection in adults, a concentration of 1.3 μg/mL of anti-capsular IgG has been used as a protection cut-off in various vaccination studies [34–36]. This threshold is serotype-specific and may be different for post-immunization and naturally acquired antibodies and may also vary with other factors including lifestyle, ethnicity, and immune status of the population in a status [34, 37]. Although anti-capsular antibody concentration and functionality in serum are used as a correlate of protection in pneumococcal vaccination studies, the protective role of pneumococcal surface protein A (PspA), cannot be ignored.

PspA, a highly heterogenous surface protein of pneumococci has been reported to be immunogenic, and antibodies generated to PspA in response to colonization protects against both carriage and invasive infections [38–41]. Based on heterogeneity, PspA can be divided into six clades which are grouped into two families. Antibodies generated to one family types of PspA cross-protects against other PspA families [42, 43]. PspA also plays an important role in the virulence of pneumococci by blocking the activation and deposition of complement on the bacterial surface, hence inhibiting phagocytosis. The protein also protects bacteria from apo-lactoferrin-mediated killing by binding to lactoferrin [38, 44–51].

The immunological response to pneumococcal vaccines is assessed by fold-increase in antibody levels relative to pre-vaccination levels [52]. Therefore, the measurement of baseline IgG titers is important for designing effective vaccine strategies. We hypothesize that an altered baseline antibody response to asymptomatic carriage will be observed in those with diabetes as compared to those with no diabetes. Given the scarcity of data on naturally acquired IgG titers to pneumococcal capsular polysaccharides (PnPs) and PspA, particularly in individuals with diabetes, the primary aim of the study is to measure baseline antibody titers to pneumococcal

capsular polysaccharide and pneumococcal surface protein A. Our second aim is to determine if the antibodies acquired by asymptomatic carriage can kill *S. pneumoniae*. To our knowledge, this is the first study conducted to measure the concentration and functionality of anti-pneumococcal antibodies in Pakistan, which is currently facing a double burden of pneumonia and diabetes.

## Methods

### Study participants

This study was approved by the Institutional Review Boards of Shalamar Hospital and Lahore University of Management Sciences. A total of 176 individuals participated ($\geq$18 years) in this study. The participants were recruited with their written consent through a convenient sampling method, from the Sakina Institute of Diabetes and Endocrinology Research (SiDER)- Shalamar Hospital from March 2017 to December 2020. A self-designed purpose developed questionnaire was used to collect sociodemographic and clinical data from the participants. Individuals who received pneumococcal vaccination and reported recent use (used in the past three months) of any antibiotics, were excluded from the study. Diabetes was diagnosed based on fasting blood glucose and glycated hemoglobulin values (HbA1c < 6.5 for diabetes and $\geq$ 6.5 for non-diabetes) as per American Diabetes Association diagnostic criteria [53].

### Sample collection

A total of 3 mL of blood was collected from each participant in clot stimulator tubes. The serum was separated by centrifugation of blood sample and transferred to sterile polypropylene freezer vials. Serum samples were stored at -80˚C until analysis.

### ELISA for measurement of anti-capsular IgG concentration

A total of 180 samples were analyzed to measure anti-capsular IgG concentrations using the World Health Organization (WHO) reference ELISA (007sp version). The detailed protocol for ELISA is available online (https://www.vaccine.uab.edu/uploads/mdocs/ELISAProtocol (007sp).pdf). Antibody concentrations were measured against four polysaccharides, namely 19F, 9V, 18C, 1. These serotypes are included in 10-valent pneumococcal conjugate vaccine (PCV10) and are responsible for 80% of infection in South Asia. Briefly, ELISA plates (Thermo Scientific™, Immulon 4HBX) were coated with four different capsular polysaccharides including 19F, 9V, 18C, 1. Each polysaccharide was mixed in an antigen coating buffer (1X PBS / 0.02% NaN3) and plates were incubated for 5 hours at 37˚C in a humidified chamber, followed by washing with wash buffer (1X TBS /0.1% Brij solution). Human serum samples and reference sera diluted and adsorbed in adsorption solution [containing 5 µg/mL of Cell wall Polysaccharide (SSI Diagnostica) and 10 µg/mL 22F pneumococcal polysaccharide (Kind gift of Dr Moon Nahm, University of Alabama)] for 30 minutes [3] were added to the ELISA plates and incubated at room temperature for 2 hours. The plates were washed 5 times with the wash buffer and anti-human IgG-labeled with alkaline phosphatase (Southern Biotech) diluted (1:2000) in antibody buffer (1x PBS, 0.02% NaN$_3$) was added to each well and incubated for 2 hours. After washing, plates were developed by adding 100 µL para nitrophenol phosphate (pNPP) (Sigma-Aldrich). The reaction was stopped by adding 50 µL of 3M NaOH to each well and plates were read at an optical density (OD) of 405–690 in a microtiter plate reader. Optical density values were converted to antibody concentrations based on the reference sera.

## Differentiation of HL60 cells into neutrophils

HL-60 cells cultured in complete RPMI [RPMI supplemented with 10% heat inactivated-FBS, 2% Glutamax (Gibco™ GlutaMax™) and 1% Penicillin-Streptomycin (Gibco™)] were centrifuged at 350$g$ for 5 minutes, supernatant was discarded, and the pellet was resuspended in HL-60 cells differentiation media (RPMI supplemented with 10% heat inactivated-FBS, 2% Glutamax (Gibco™ GlutaMax™) and 1.27% Dimethyl Sulfoxide (DMSO), counted, cell number adjusted to 4 x 10$^5$ cells/ml, and 70 mL cell suspension was added to 150 cm$^2$ culture flask. The cells were then incubated in tissue culture incubator (37˚C, 5% CO$_2$) for four days to differentiate into neutrophil-like cells.

## Multiplexed opsonophagocytic killing assay for measuring the functionality of antibodies

To evaluate the functionality of naturally acquired anti-capsular polysaccharide we performed Multiplexed opsonophagocytic killing assay (MOPA) using the established protocol detailed in reference [54]. Briefly, opsonophagocytosis was measured as follows; Heat inactivated human sera were serially diluted 3-fold in a microtiter plate (initial serum dilution 1:5), followed by the addition of antibiotic resistant strains of pneumococcal serotype 19F, 18C, and 9V. Plates were incubated for 30 minutes on a shaker (750 RPM) at room temperature, followed by the addition of 10 µL baby rabbit complement (Pel-freeze biologicals) and differentiated HL-60 cells (cells: bacteria was 200:1). Plates were incubated for another 45 minutes at 37˚C in a humidified chamber, followed by cooling on ice for 15 minutes to stop opsonophagocytosis. At the end of all incubations, a 10-µl volume of mixture was spotted from each well onto blood agar plates containing different antibiotics against resistant pneumococcal strains used in the assay. Blood agar plates were incubated overnight, and bacterial colonies were counted. Opsonic titer, defined as the interpolated reciprocal serum dilution that resulted in the killing of 50% assay bacteria, was determined using Opsotiter3 software (provided by Robert Burton; University of Alabama).

## Measurement of serum antibodies concentration to PspA

Immunoglobulin G (IgG) to PspA were measured in 90 samples using standard Enzyme-linked immunosorbent assay (ELISA) [12, 55]. Immuno Nonsterile 96-Well Plates (Thermo Scientific™, Immulon 4HBX) were coated with 10 µg/ml of Family 2 PspA (the most commonly found in bacteria causing invasive infections [56]) in coating buffer (25 ml 0.06 M Na$_2$HCO$_3$, 20 ml 0.06M Na$_2$CO$_3$, 5 ml deionized H$_2$O, pH 9.6) at 4˚C overnight, followed by blocking for 1 h at room temperature using phosphate-buffered saline (PBS) containing 1% bovine serum albumin (BSA Sigma-Aldrich, CA). To account for non-specific binding of serum to BSA protein, plates coated with 1% BSA only were used as a negative control. Serum samples were diluted 1:100 in PBS-1% BSA and added to ELISA plates. The plates were incubated at 37˚C for 1 hour and washed 3 times with PBST (PBS containing 0.05% Tween 20). A goat anti-human alkaline phosphatase-conjugated antibody (Southern Biotech) diluted 1:1000 was added to the plates and incubated for 1 hour at room temperature. Plates were washed 3 times with PBST and developed using 100ul of para nitrophenyl phosphate (pNPP) (Sigma-Aldrich). The absorbance was read at 450nm using a Synergy-HTX multimode plate reader.

## Statistical analysis

Statistical analysis was performed on GraphPad Prism 8.0.2. The antibody concentration and opsonic titer data were found to be not normally distributed and skewed. Therefore, non-

parametric tests were used for the analysis of the data. The antibody concentration and opsonic titers were reported as geometric mean concentration/titer with 95% CI. For the comparison of two independent groups, Mann-Whitney U, and more than two groups Kruskal Wallis test was used. To explore the association between the continuous independent variables and outcomes, a non-parametric Spearman correlation ($r$) test was used. A p-value of $< 0.05$ was considered statistically significant for all analyses.

## Results

### Characteristics of the participants

A total of 176 adults participated in this study. The median age (years) of these participants was 47 (40–52). Females represented 52.3% (n = 92) while males 47.7% (n = 84) of the total recruited participants ($p$ = 0.546). Of these 176 participants, 124 (70.5%) were individuals with type 2 diabetes and 52 (29.5%) were without diabetes. The demographic and clinical data of those with and without type 2 diabetes are listed in **Table 1**.

### Measurement of naturally acquired serotype-specific anti-pneumococcal polysaccharide IgG (anti-PnPs IgG)

Serotype-specific anti-capsular IgG concentrations were measured against 4 pneumococcal capsular polysaccharides (PnPs), namely, 19F, 1, 9V and 18C. Pakistan introduced PCV10 into its Expanded Program on Immunization (EPI) in 2012. Although vaccination can result in herd immunity and lowering the disease burden, a recent publication from Pakistan shows that serotypes included in PCV10 are still among the most prevalent serotypes and are responsible for high carriages and high disease burden [57]. Additionally, cross-protection against different polysaccharides is rarely achieved, however, a robust cross-protection against proteins antigens is observed more frequently. Among the 4 serotypes, the highest IgG concentration was observed against 19F (GMC: 4.213, 95% CI: 3.756–4.725), followed by 18C (GMC: 3.403, 95% CI: 3.108–3.727), 9V (GMC: 1.899, 95% CI: 1.692–2.109) and serotype 1 (GMC: 1.027, 95% CI: 0.929–1.135). The geometric mean IgG concentrations with 95% confidence intervals, for all 4 serotypes are shown in **Fig 1**.

To explore the effects of diabetes status on the production of anti-capsular IgG concentrations, we divided participants into two groups based on their blood glucose control, the groups included (i) those with type 2 diabetes (HbA1c $\geq$ 6.5%) and (ii) those without type 2 diabetes

**Table 1. Demographic and clinical data of participants.** The data is represented as median (interquartile range) and mean (standard deviation) for continuous variables, and as percentages for categorical variables. The Mann-Whitney U and t-test were used for the comparison of the two groups. A p-value < 0.05 was considered statistically significant.

| Characteristics | Diabetes participants | Non-diabetes participants | *p*-value |
|---|---|---|---|
| **Age (years)** | 49 (10.0) | 46 (4.0) | <0.001 |
| **No. (%male)** | 54 (43.54%) | 30 (57.692%) | 0.087 |
| **BMI (Kg/m$^2$)** | 30.785 (11.060) | 26.560 (3.830) | <0.001 |
| **HbA1c (%)** | 9.450 (2.650) | 5.70 (0.400) | <0.001 |
| **LDL (mg/dL)** | 124.333 (36.421) | 102.818 (35.740) | 0.041 |
| **HDL (mg/dL)** | 39.280 (8.434) | 41.818 (12.139) | 0.357 |
| **TG (mg/dL)** | 171.0 (170.50) | 119.0 (151.00) | 0.133 |
| **Cholesterol (mg/dL)** | 190.00 (196.0) | 155.0 (47.00) | 0.031 |
| **FHD** | 92 (74.796%) | 18 (40.909%) | <0.0001 |

LDL, Low-density lipoprotein; HDL, High-density lipoprotein; TG, Triglyceride; FHD, Family history of diabetes.

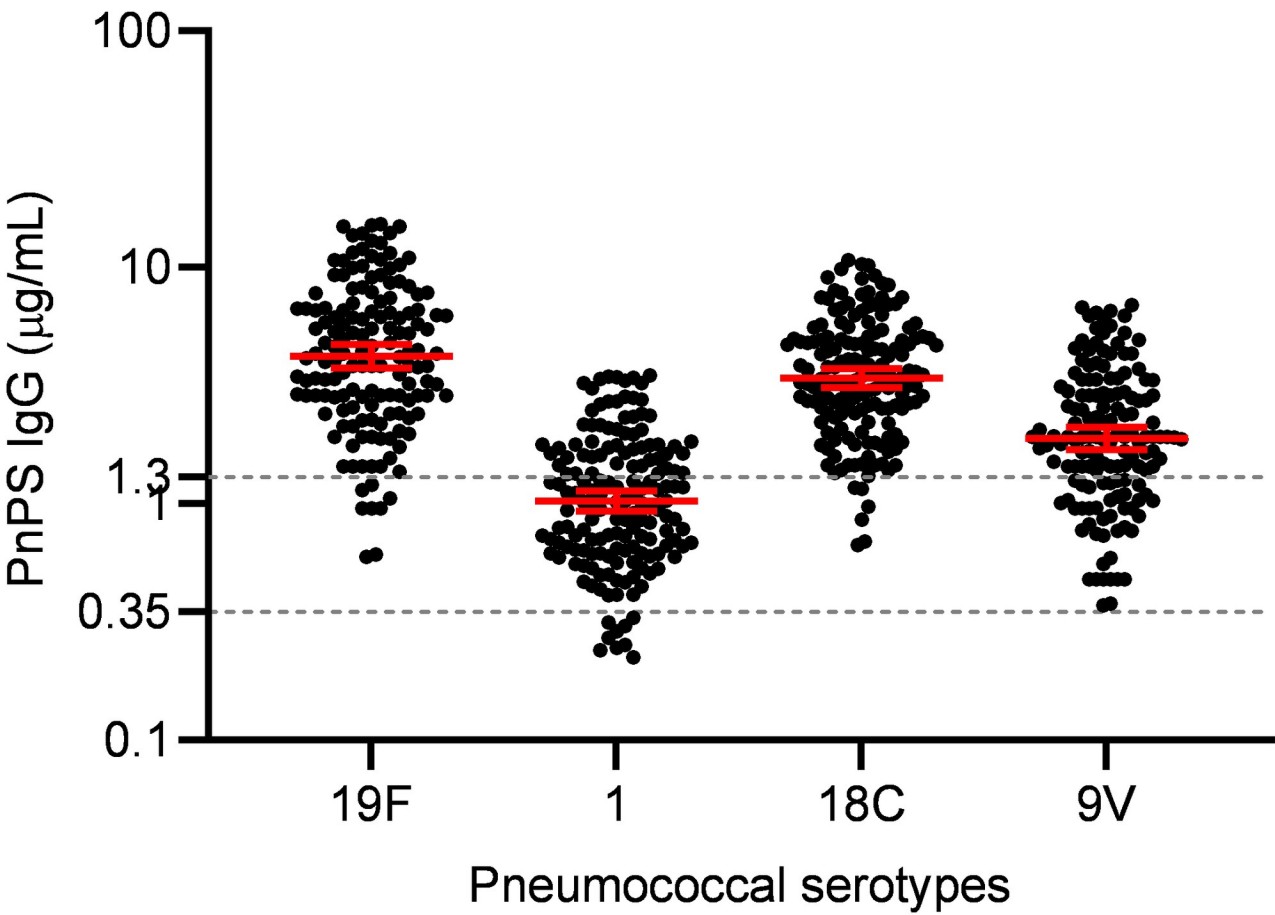

**Fig 1. Serotype-specific anti-capsular IgG concentration against serotype 1, 9V, 18C and 19F.** Serotype-specific anti-capsular IgG concentration was measured using WHO 007SP reference ELISA. Kruskal Wallis test with Dunn's multiple comparison post hoc was used for comparison of the IgG concentration against different serotypes. Anti-capsular IgG concentrations are shown on a logarithmic scale (log10) with the geometric mean and 95% confidence interval. The dotted lines represent protective thresholds, i.e., 0.35 μg/mL and 1.3 μg/mL.

(HbA1c < 6.5%). On comparison, no significant difference was observed in anti-capsular IgG concentrations for any of the 4 serotypes between the two groups (**Fig 2**).

When categorizing the participants into two groups based on their BMI (i) Obese; BMI $\geq$ 30 kg/m$^2$, and (ii) Non-Obese BMI: < 30 kg/m$^2$), we found that non-obese individuals have a higher anti-capsular IgG concentration compared to obese individuals, against serotype 1, 9V and 18C (**Fig 3**).

We observed no significant difference in anti-capsular IgG concentration between males and females for serotypes 1, 18C, and 9V, but for 19F, females showed a significantly higher anti-capsular IgG concentration compared to males ($p$ = 0.020). Age plays a significant role in determining immune response to polysaccharide antigens. It has been shown that children and elderly do not respond to polysaccharide antigens owing to underdeveloped immune system and immune senescence. Therefore, to determine whether antibody concentration to anti-capsular varies with age, we categorized the participants into two groups, young adults (age < 45 years) and older adults (age $\geq$ 45 years). On comparison, we found no significant difference in IgG concentration between the two age groups for any of the 4 serotypes. To determine whether the use of insulin among diabetes participants affects the production of anti-capsular IgG, we compared those using insulin with those not using insulin among

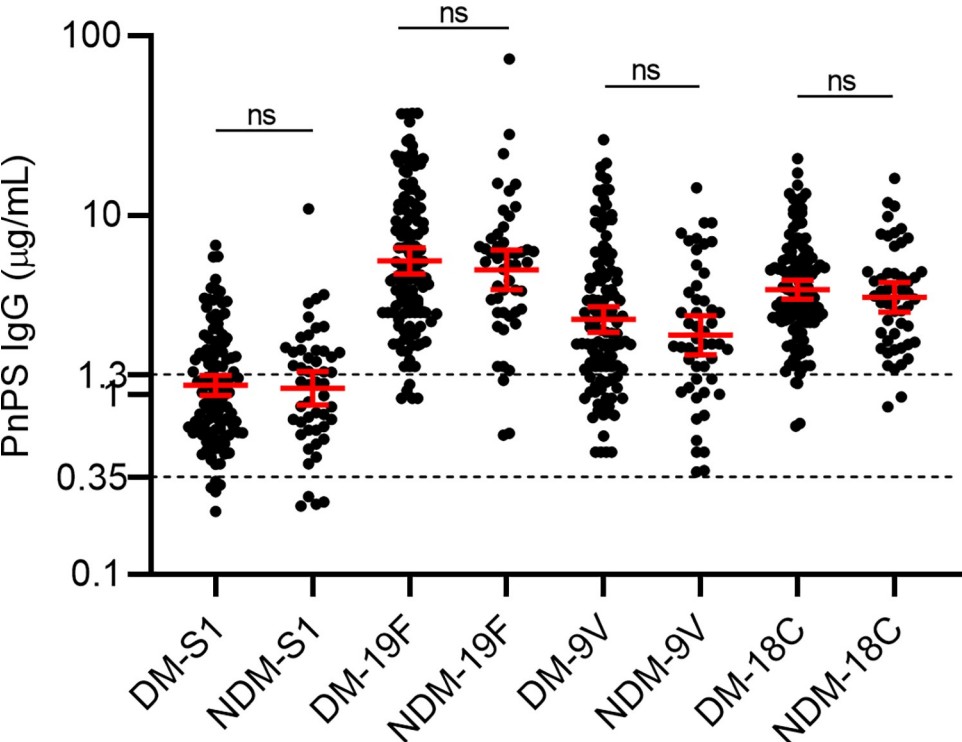

**Fig 2. Serotype-specific anti-capsular IgG concentration in individuals with (DM) and without type 2 diabetes (NDM).** Serotype-specific anti-capsular IgG concentration was measured for serotype 1, 19F, 18C and 9V using WHO 007 reference ELISA. Mann-Whitney U test was used to compare diabetes and non-diabetes group for each serotype. Antibody concentrations are shown on a logarithmic scale (log10) with the geometric mean and 95% confidence interval. The dotted lines represent protective thresholds, i.e., 0.35 μg/mL and 1.3 μg/mL. A p-value < 0.05 was considered statistically significant.

diabetes individuals. We found no significant difference in anti-capsular IgG concentrations between the two groups for any of 4 the serotypes. Also, no significant difference was observed in the anti-capsular IgG concentration in those with and without a family history of type 2 diabetes, and those who were using anti-diabetic drugs (ADD) in combination with insulin compared to those who were using only anti-diabetic drugs (**Table 2**).

When BMI was stratified by diabetes status, we found elevated IgG concentrations in non-obese diabetes individuals compared to obese diabetes individuals for serotypes 1, 9V, and 18C. Also, we found no significant difference when age was stratified by diabetes status (**Table 3**). We analyzed the correlation of age, BMI, and HbA1c with the production of anti-capsular IgG and found that only BMI was significantly negatively correlated (r = -0.202, p = 0.014) with anti-pneumococcal serotype 1 antibodies. There were no significant correlations between any other variables with antibody concentration against any of the four serotypes (**Fig 4**). When we analyzed correlation between BMI and anti-PnPs IgG in diabetes and non-diabetes groups individually, we found a moderately significant negative correlation only between anti-PnPs1 IgG and BMI in the diabetes group (r = -0.193, p = 0.048). No significant correlation was found between anti-PnPs IgG and BMI in the non-diabetes group.

## Measurement of opsonic titers (OPA) for serotypes 19F, 9V and 18C

To determine the functionality of serotype-specific anti-pneumococcal antibodies, we performed multiplexed opsonophagocytic killing assay on 39 samples (24 with diabetes and fifteen

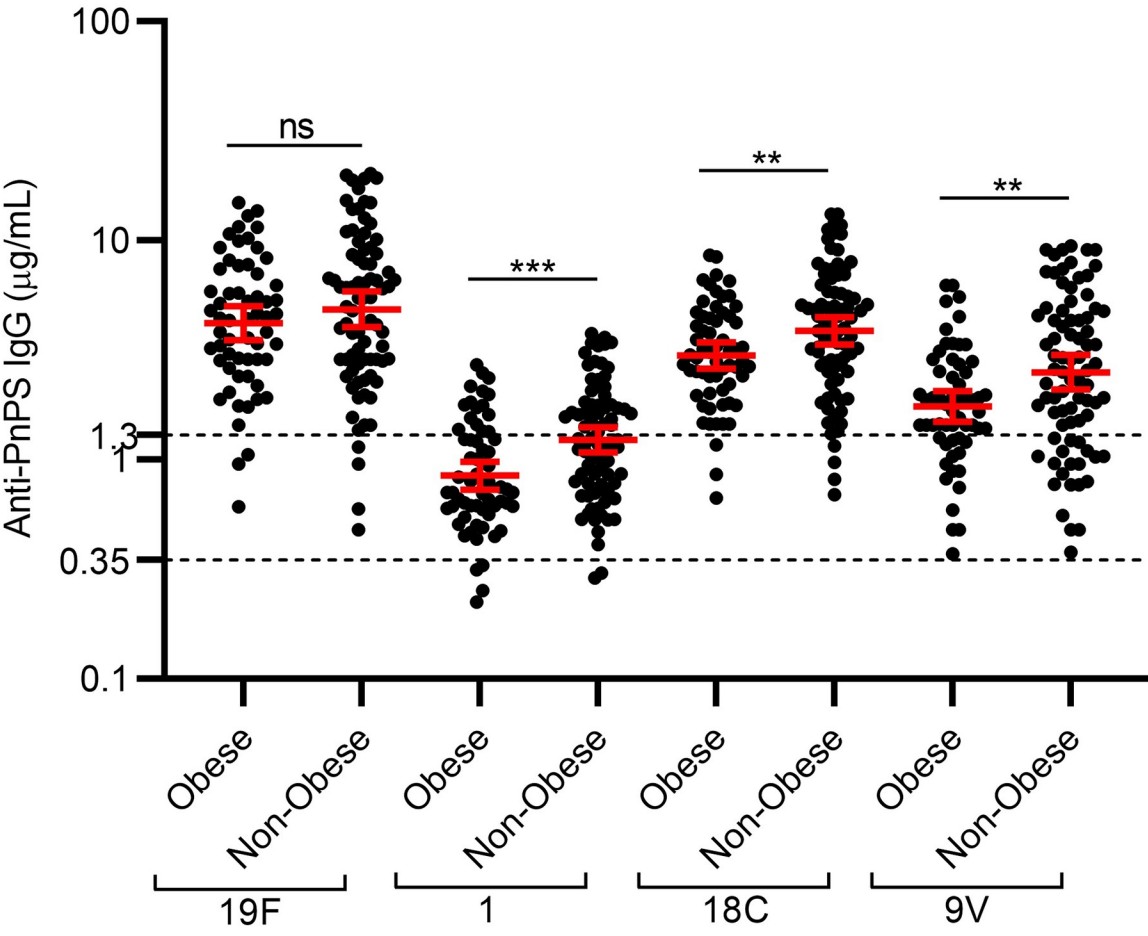

**Fig 3. Comparison of serotype-specific anti-capsular IgG concentrations in obese and non-obese individuals.** The serotype-specific anti-capsular IgG concentration between obese and non-obese groups were compared using Mann-Whitney U test. Antibody concentrations are shown on a logarithmic scale (log10) with the geometric mean and 95% confidence interval. The dotted lines represent protective thresholds, i.e., 0.35 μg/mL and 1.3 μg/mL. A p-value < 0.05 was considered statistically significant.

without diabetes) for serotypes 19F, 9V, and 18C. Serum samples with opsonic titers < 1:5 (initial serum dilution) was given a value of 2 for analysis purposes. While there are no established protective opsonic titers in adults, for children, vaccination that achieves opsonic titer of ≥ 1:8 is considered protective. The opsonic titer threshold of ≥ 1:8 was only observed for 18C (GMT: 13.97, 95% CI: 9.749–20.03). For 19F (GMT: 7.890, 95% CI: 5.518–11.28) and 9V (GMT: 7.026, 95% CI: 4.732–10.43) the geometric mean opsonic titers were < 1:8 (**Fig 5**). The percentages of individuals achieving opsonic titers ≥ 1:8 was 43.89% (n = 17) for serotype 19F, 41.02% (n = 16) for 9V and 76.92% (n = 30) for 18C.

When compared by gender, no significant difference in the opsonic titers between males and females was observed for any of the 3 serotypes. When categorized into obese and non-obese groups, the difference in opsonic titers remains non-significant between the two groups for all the serotypes. On comparison of the opsonic titers between those with (HbA1c ≥ 6.5%) and without diabetes (HbA1c < 6.5%), we found that individuals without diabetes have significantly higher opsonic titers compared to individuals with diabetes for 19F and 9V. Opsonic titers were comparable between the two groups for serotype 18C (**Fig 6 and Table 4**). For 19F, 41.66% (n = 10) showed opsonic titers ≥ 1:8 in the diabetes group as compared to 66.66%

**Table 2. Represents geometric mean IgG concentration with a 95% confidence interval against different pneumococcal capsular polysaccharides (PnPs 19F, PnPs 1, PnPs 18C, and PnPs 9V) based on participants' characteristics.** The Mann-Whitney U test was used to compare the antibody concentration between the two groups. A p-value < 0.05 was considered significant.

| Subject Characteristics | Serotype-specific IgG GMC (95% CI) | | | |
|---|---|---|---|---|
| | Serotype 19F | Serotype1 | Serotype 18C | Serotype 9V |
| **Diabetes Status** | | | | |
| Diabetes | 5.567 (4.715–6.574) | 1.131 (0.993–1.287) | 3.862 (3.421–4.361) | 2.636 (2.238–3.104) |
| No Diabetes | 4.962 (3.857–6.383) | 1.090 (0.878–1.353) | 3.499 (2.899–4.223) | 2.153 (1.675–2.766) |
| | *p*-value = 0.6531 | *p*-value = 0.9436 | *p*-value = 0.3789 | *p*-value = 0.2807 |
| **Gender** | | | | |
| Male | 4.541(3.749–5.502) | 1.052 (0.8797–1.259) | 3.462 (3.001–3.992) | 2.298 (1.887–2.799) |
| Female | 6.193 (5.113–7.500) | 1.146 (0.9788–1.342) | 4.075 (3.538–4.694) | 2.633 (2.181–3.178) |
| | *p*-value = 0.020 | *p*-value = 0.644 | *p*-value = 0.122 | *p*-value = 0.334 |
| **Insulin use among diabetes** | | | | |
| Insulin | 4.733 (3.505–6.392) | 1.038 (0.8407–1.283) | 3.507 (2.777–4.431) | 2.843 (2.016–4.009) |
| No Insulin | 6.005 (4.913–7.338) | 1.161(0.9837–1.371) | 4.038 (3.484–4.680) | 2.578 (2.127–3.125) |
| | *p*-value = 0.191 | *p*-value = 0.393 | *p*-value = 0.702 | *p*-value = 0.580 |
| **Family History of Diabetes** | | | | |
| Yes | 5.664 (4.796–6.688) | 1.192 (1.036–1.371) | 4.006 (3.502–4.583) | 2.652 (2.233–3.149) |
| No | 4.782 (3.678–6.217) | 0.964 (0.795–1.169) | 3.396 (2.894–3.985) | 2.228 (1.735–2.861) |
| | *p*-value = 0.232 | *p*-value = 0.087 | *p*-value = 0.092 | *p*-value = 0.181 |
| **Body Mass Index (BMI)** | | | | |
| Obese (BMI $\geq$ 30 Kg/m$^{2)}$) | 4.202 (3.513–5.025) | 0.847 (0.731–0.980) | 2.991 (2.608–3.430) | 1.753 (1.494–2.058) |
| Non-Obese (BMI < 30 Kg/m$^{2)}$) | 4.863 (4.037–5.859) | 1.233 (1.081–1.406) | 3.878 (3.362–4.472) | 2.510 (2.095–3.007) |
| | *p*-value = 0.242 | *p*-value = 0.003 | *p*-value = 0.008 | *p*-value = 0.005 |
| **Age** | | | | |
| Age $\geq$ 45 | 3.960 (3.385–4.631) | 1.004 (0.881–1.144) | 3.559 (3.158–4.011) | 1.922 (1.667–2.215) |
| Age < 45 | 4.681 (3.933–5.571) | 1.085 (0.927–1.270) | 3.284 (2.829–3.811) | 2.074 (1.732–2.483) |
| | *p*-value = 0.187 | *p*-value = 0.425 | *p*-value = 0.505 | *p*-value = 0.507 |

(n = 10) in the non-diabetes group. It was 29.6% (n = 7) vs 66.66% (n = 10) for 9V and 70.83% (n = 17) vs 80% (n = 12) for 18C in diabetes and non-diabetes groups respectively.

When diabetes was stratified by BMI, we observed elevated opsonic titers in non-obese, non-diabetes individuals compared to their non-obese diabetes counterparts. However, no significant difference in opsonic titers was identified between obese diabetes and obese non-diabetes participants (**Table 5**).

Although correlation analysis showed that OPA titers were positively correlated with anti-pneumococcal IgG concentrations for all the 3 serotypes, the association was found to be statistically significant only for 19F (**Fig 7**).

When OPA titers were analyzed for association with HBA1c, age, and BMI, we found that HbA1c had a statistically significant negative correlation with OPA titers for 19F and 9V (**Fig 8**).

## Measurement of anti-pneumococcal surface protein-A IgG (anti-PspA IgG)

We measured anti-PspA IgG in 90 samples (68 diabetes and 22 non-diabetes). The mean age (years) of these participants was 44.27 (11.94). Antibodies to PspA protein were measured as described in materials and methods. In the absence of a quantifiable standard, comparison withing diabetes and non-diabetes groups were made using geometric mean of the optical density for IgG. The optical density for samples was adjusted by subtracting serum values from

**Table 3. Serotype-specific anti-PnPs IgG concentration after stratification of diabetes status by BMI and age, and then stratification of BMI and age by diabetes status.** The IgG concentration is reported as Geometric mean concentration (GMC) with a 95% confidence interval (95% CI). For the comparison of antibody concentration between the two groups, the Mann-Whitney U test was used and a p-value < 0.05 was considered significant.

| Stratification group and characteristics | | Serotype 19F | | Serotype 1 | | Serotype 18C | | Serotype 9V | |
|---|---|---|---|---|---|---|---|---|---|
| | | IgG GMC (95% CI) | P-value | IgG GMC (95% CI) | P-value | IgG GMC (95% CI) | P-value | IgG GMC (95% CI) | P-value |
| Participants with BMI ≥ 30 (Kg/m²) | Diabetes | 4.106 (3.412–4.940) | 0.211 | 0.854 (0.739–0.987) | 0.621 | 3.140 (2.705–3.645) | 0.409 | 1.757 (1.477–2.090) | 0.895 |
| | Non-diabetes | 5.024 (2.597–9.720) | | 0.942 (0.537–1.652) | | 2.743 (1.725–4.360) | | 1.740 (0.984–3.075) | |
| Participants with BMI < 30 (Kg/m²) | Diabetes | 5.553 (4.342–7.101) | 0.246 | 1.186 (0.940–1.496) | 0.736 | 4.239 (3.522–5.103) | 0.049 | 4.239 (3.522–5.103) | 0.709 |
| | Non-diabetes | 4.295 (3.256–5.664) | | 1.192 (0.979–1.450) | | 3.173 (2.544–3.957) | | 2.360 (1.667–3.341) | |
| Participants with age ≥ 45 | Diabetes | 4.332 (3.603–5.209) | 0.814 | 0.971 (0.817–1.154) | 0.397 | 3.524 (3.083–4.027) | 0.763 | 1.908 (1.631–2.231) | 0.731 |
| | Non-diabetes | 3.839 (2.725–5.408) | | 0.870 (0.642–1.180) | | 3.363 (2.549–4.438) | | 1.714 (1.151–2.552) | |
| Participants with age < 45 | Diabetes | 4.607 (3.656–5.806) | 0.600 | 0.995 (0.818–1.215) | 0.178 | 3.782 (3.026–4.726) | 0.202 | 2.156 (1.697–2.738) | 0.456 |
| | Non-diabetes | 5.070 (3.877–6.630) | | 1.165 (0.899–1.510) | | 3.064 (2.465–3.807) | | 1.947 (1.481–2.560) | |
| Participants with diabetes | BMI ≥ 30 (Kg/m²) | 4.106 (3.412–4.940 | 0.091 | 0.854 (0.739–0.987) | 0.003 | 3.140 (2.705–3.645) | 0.010 | 1.757 (1.477–2.090) | 0.008 |
| | BMI < 30 (Kg/m²) | 5.553 (4.342–7.101) | | 1.186 (0.940–1.496) | | 4.239 (3.522–5.103) | | 4.239 (3.522–5.103) | |
| No diabetes | BMI ≥ 30 (Kg/m²) | 5.024 (2.597–9.720) | 0.404 | 0.942 (0.537–1.652) | 0.350 | 2.743 (1.725–4.360) | 0.601 | 1.740 (0.984–3.075) | 0.379 |
| | BMI < 30 (Kg/m²) | 4.295 (3.256–5.664) | | 1.192 (0.979–1.450) | | 3.173 (2.544–3.957) | | 2.360 (1.667–3.341) | |
| Participants with diabetes | Age ≥ 45 | 4.332 (3.603–5.209) | 0.581 | 0.971 (0.817–1.154) | 0.913 | 3.524 (3.083–4.027) | 0.478 | 1.908 (1.631–2.231) | 0.399 |
| | Age < 45 | 4.607 (3.656–5.806) | | 0.995 (0.818–1.215) | | 3.782 (3.026–4.726) | | 2.156 (1.697–2.738) | |
| No diabetes | Age ≥ 45 | 3.839 (2.725–5.408) | 0.348 | 0.870 (0.642–1.180) | 0.079 | 3.363 (2.549–4.438) | 0.619 | 1.714 (1.151–2.552) | 0.826 |
| | Age < 45 | 5.070 (3.877–6.630) | | 1.165 (0.899–1.510) | | 3.064 (2.465–3.807) | | 1.947 (1.481–2.560) | |

reagent control and controls lacking antigen or secondary antibody. The residual optical density was considered as a positive value for IgG. Females represented 51.6% of the participants. Individuals with and without diabetes demonstrated comparable anti-PspA IgG (GM: 0.550, 95% CI: 0.474–0.638 (GM: 0.485, CI: 0.368–0.638) (p = 0.409) (Fig 9). Anti-PspA IgG, were also comparable between females (GM: 0.566, 95% CI: 0.466–0.687) and males (GM: 0.501, 95% CI: 0.420–0.596), (p = 0.343).

To further explore the effects of gender and diabetes on the production of anti-PspA IgG, diabetes was stratified by gender. We found that females with diabetes have comparable IgG (GM: 0.609, 95% CI: 0.495–0.750) to males with diabetes (GM: 0.478, 95% CI: 0.386–0.592), (p = 0.134). Also, no statistically significant difference in the IgG was observed between non-diabetes males (GM: 0.546, 95% CI: 0.390–0.765) and non-diabetes females (GM: 0.374, 95% CI: 0.212–0.659), (p = 0.298). We also found that females with diabetes have comparable IgG to females without diabetes (p = 0.081). There were no statistically significant differences observed in anti-PspA IgG among individuals with (GM: 0.551, 95% CI: 0.467–0.650) or

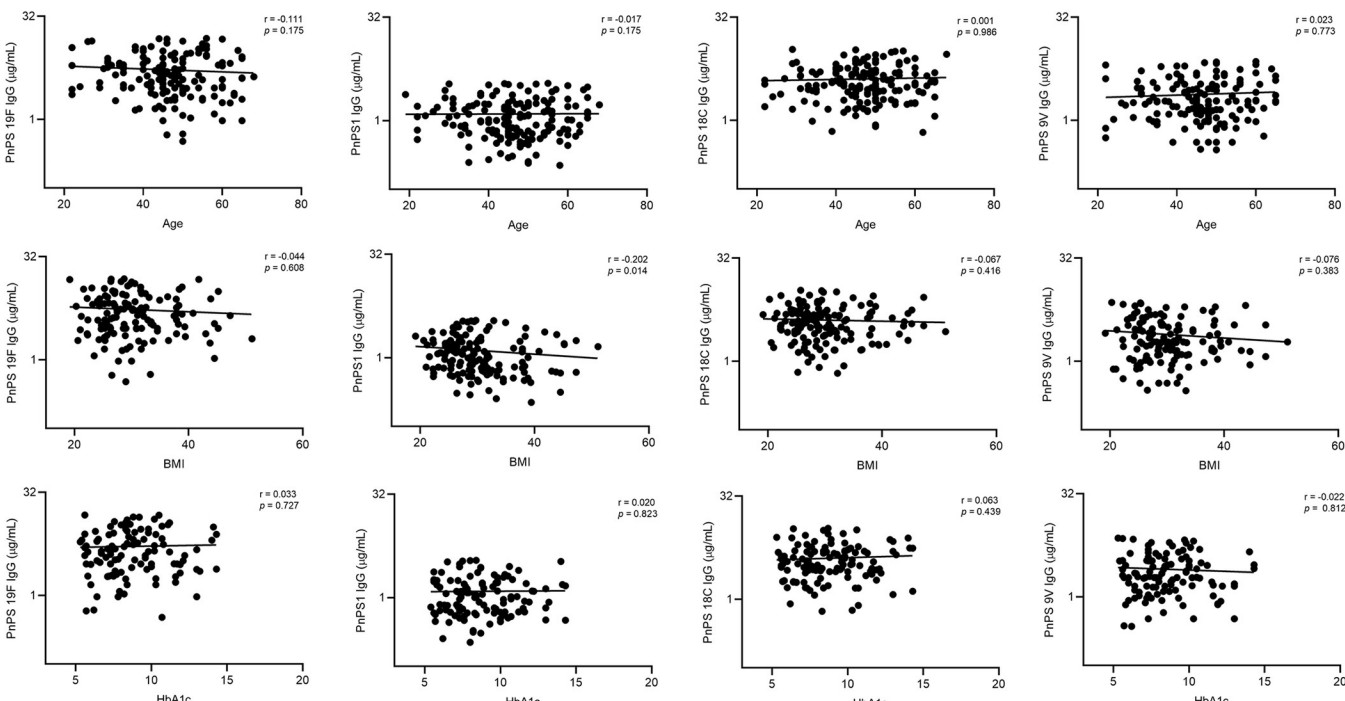

**Fig 4. Association of serotype-specific anti-capsular IgG concentration with HbA1c, BMI, and Age.** Spearman correlation test (r) was used to analyze the data. A p-value < 0.05 was considered statistically significant. The only significant association was observed between antiPnps1 IgG and BMI.

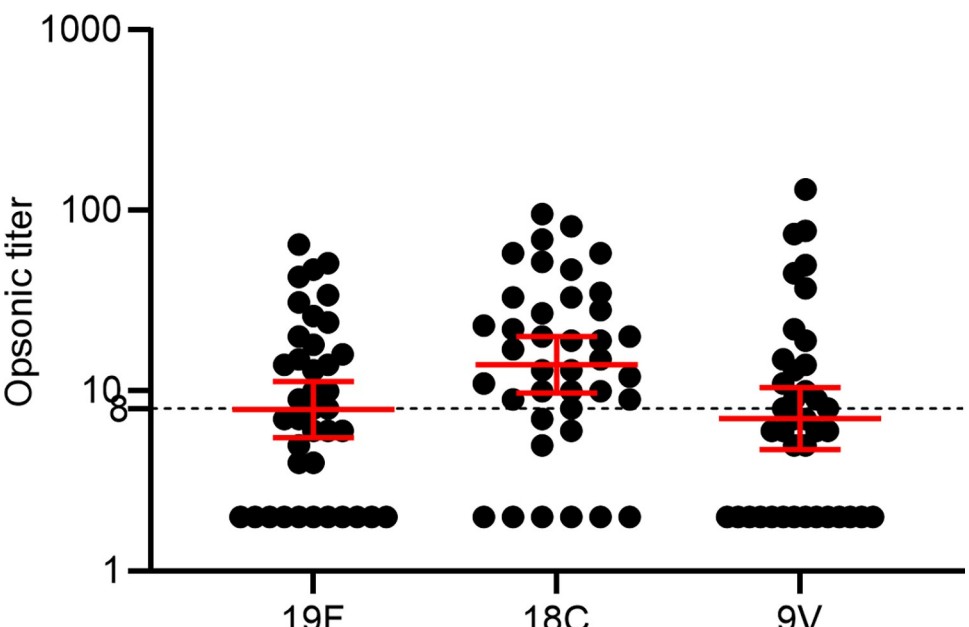

**Fig 5. Serotype-specific opsonic titers for 19F, 18C and 9V.** Opsonic titers were measured against serotype 19F, 18C and 9V using multiplexed opsonophagocytic killing assay (MOPA). Opsonic titers are shown on a logarithmic scale (log10) with the geometric mean and 95% confidence interval. The dotted line at 8 represents the cut-off for protective opsonic titer.

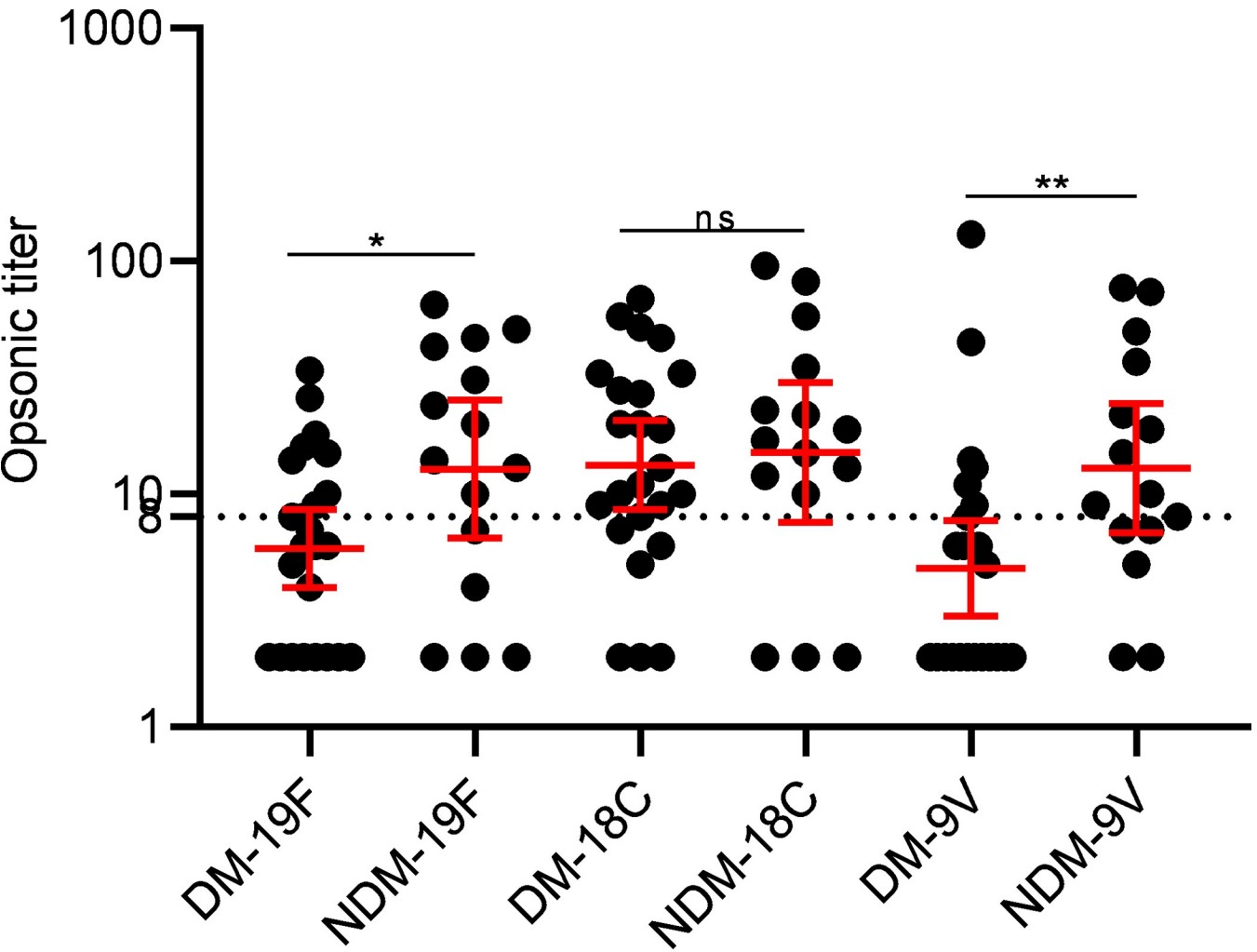

**Fig 6. Serotype-specific opsonic titers in individuals with (DM) and without type 2 diabetes (NDM).** Opsonic titers were measured in individuals with and without type 2 diabetes against serotype 19F, 9V and 18C using multiplexed opsonophagocytic killing assay (MOPA). Mann-Whitney U test was used for comparison between the two groups. Opsonic titers are shown on a logarithmic scale (log10) with the geometric mean and 95% confidence interval. The dotted line at 8 represents the cut-off for protective opsonic titer. A p-value < 0.05 was considered statistically significant.

without (GM: 0.534, 95% CI: 0.418–0.681) a family history of diabetes (p = 0.682), as well as between individuals among diabetes who were using insulin (GM: 0.496, 95% CI: 0.360–0.683), and those who were not (GM: 0.589, 95% CI: 0.492–0.705), (p = 0.277), and between obese (GM: 0.479, 95% CI: 0.384–0.598) and non-obese individuals (GM: 0.568, 95% CI: 0.484–0.667), (p = 0.086). The anti-PspA IgG were comparable between individuals who were using anti-diabetic drugs in combination with insulin compared to those who were using only insulin (**Fig 10**).

On stratification of diabetes status by BMI, we found that obese diabetes individuals (GM: 0.497, 95% CI: 0.393–0.629) have comparable IgG with those of obese non-diabetes (GM: 0.334, 95% CI: 0.094–1.178). Also, non-obese diabetes (GM: 0.612, 95% CI: 0.503–0.743) were found to have comparable IgG with those of non-obese non-diabetes (GM: 0.514, 95% CI: 0.380–0.695).

To find the association of anti-PspA IgG with age, BMI, and HbA1c, we used Spearman correlation analysis. A statistically non-significant but negative correlation for age (r = -0.160,

**Table 4. Represents geometric mean opsonic titers with 95% confidence intervals against serotypes 19F, 9V, and 18C based on participants' characteristics.** The Mann-Whitney U test was used to compare the opsonic titers between the two groups. A p-value < 0.05 was considered significant.

| Subject Characteristics | Geometric mean opsonic titers (95%CI) | | |
|---|---|---|---|
| | Serotype 19F | Serotype 9V | Serotype 18C |
| **Diabetes Status** | | | |
| Diabetes | 5.831 (3.959–8.587) | 4.802 (2.993–7.703) | 13.31 (8.566–20.67) |
| No Diabetes | 12.80 (6.475–25.31) | 12.920 (6.816–24.49) | 15.11 (7.584–30.11) |
| | p = 0.037 | p = 0.006 | p = 0.552 |
| **Gender** | | | |
| Male | 8.351 (4.739–14.72) | 8.140 (4.438–14.93) | 17.80 (10.00–31.69) |
| Female | 7.432 (4.573–12.08) | 6.018 (3.461–10.46) | 10.83 (6.912–16.97) |
| | p = 0.838 | p = 0.488 | p = 0.162 |
| **BMI** | | | |
| Obese | 7.655 (4.818–12.16) | 6.116 (3.646–10.26) | 12.31 (6.810–22.24) |
| Non-Obese | 8.077 (4.623–14.11) | 7.821 (4.249–14.40) | 15.42 (9.486–25.05) |
| | p = 0.999 | p = 0.746 | p = 0.550 |

p = 0.132) and BMI (r = -0.125, p = 0.239), and a positive correlation for HbA1c (r = 0.029, p = 0.813) was observed with anti-PspA IgG (**Fig 11**).

## Discussion

Protection from both carriage and invasive pneumococcal infection is dependent upon the production of antibodies with protective titers that induce opsonophagocytosis of *S. pneumoniae* [55, 58, 59]. The production of antibodies, in response to carriage, infection, and or vaccination, and their protective ability is dependent upon various factors including the nature of the pathogen and the health of an individual [60]. Individuals with diabetes are highly susceptible to respiratory tract infections as compared to non-diabetes individuals. Pneumococcal infections are severe and often fatal in those with diabetes [61]. To determine if the susceptibility of individuals with diabetes stems from low concentrations or poorly protective capacity of antibodies, we assessed both antibody titers and their protective functions.

The comparable IgG concentrations observed among individuals with and without diabetes in our study align with previous research findings where antibodies were measured to vaccines against other bacterial and viral infections in those with and without diabetes. Mathews et al., found comparable anti-capsular IgG concentrations in Mexican American adults with and

**Table 5. Serotype-specific opsonic titers after stratification of diabetes status by BMI, and then stratification of BMI by diabetes status.** For the comparison of opsonic titers between the two groups, the Mann-Whitney U test was used and a p-value < 0.05 was considered significant. The opsonic titer is a geometric mean (GMT) with a 95% confidence interval (95% CI).

| Stratification group and characteristics | | Serotype 19F | | Serotype 9V | | Serotype 18C | |
|---|---|---|---|---|---|---|---|
| | | OPA GMT (95%CI) | P-value | OPA GMT (95%CI) | P-value | OPA GMT (95%CI) | P-value |
| **Participants with BMI ≥ 30 (Kg/m²)** | Diabetes | 7.311 (4.272–12.51) | 0.5634 | 6.032 (3.403–10.69) | 0.843 | 15.45 (8.132–29.34) | 0.244 |
| | Non-diabetes | 8.888 (1.646–48.00) | | 6.395 (0.754–54.23) | | 5.881 (0.783–44.16) | |
| **Participants with BMI < 30 (Kg/m²)** | Diabetes | 4.463 (2.411–8.259) | 0.029 | 3.666 (1.544–8.708) | 0.001 | 11.16 (5.538–22.48) | 0.136 |
| | Non-diabetes | 14.62 (6.070–35.20) | | 16.68 (8.368–33.27) | | 21.30 (10.20–44.48) | |
| **Participants with diabetes** | BMI ≥ 30 (Kg/m²) | 7.311 (4.272–12.51) | 0.163 | 6.032 (3.403–10.69) | 0.084 | 15.45 (8.132–29.34) | 0.597 |
| | BMI < 30 (Kg/m²) | 4.463 (2.411–8.259) | | 3.666 (1.544–8.708) | | 11.16 (5.538–22.48) | |
| **No diabetes** | BMI ≥ 30 (Kg/m²) | 8.888 (1.646–48.00) | 0.409 | 6.395 (0.754–54.23) | 0.326 | 5.881 (0.783–44.16) | 0.175 |
| | BMI < 30 (Kg/m²) | 14.62 (6.070–35.20) | | 16.68 (8.368–33.27) | | 21.30 (10.20–44.48) | |

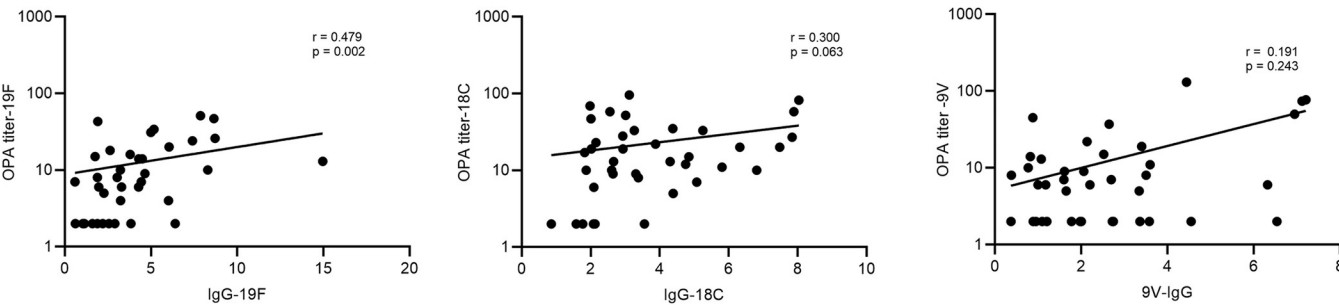

**Fig 7. Correlation analysis between serotype-specific opsonic titer and anti-pneumococcal capsular polysaccharide IgG.** Spearman (r) test was used to find a correlation between anti-capsular IgG and opsonic titers. Although a positive correlation was observed between IgG and opsonic titer for all 3 serotypes, only the correlation between IgG against 19F capsular polysaccharide and 19F opsonic titer was statistically significant (r = 0.479; p < 0.002).

without type 2 diabetes [12]. Similarly, Lederman et al., reported comparable anti-capsular IgG concentrations in both populations vaccinated with pneumococcal polysaccharides [62]. Pozzilli et al., demonstrated comparable IgG titers in individuals with and without type 2 diabetes immunized with influenza vaccine [63]. In addition to antibody titer, cellular responses were also measured to vaccine in diabetes and non-diabetes individuals and were found to be comparable. Previous reports, by Daniela et al., have highlighted comparable plasma B cell populations and functions between these groups, potentially explaining the observed IgG concentration parity [64]. Furthermore, the use of the first-line anti-diabetic drug, Metformin, is suggested as a contributing factor to the improved immune responses [65, 66]. Diaz et al, showed improved B cell functionality and antibody response to the influenza vaccine in individuals with type 2 diabetes using Metformin [67].

The documented elevated anti-capsular IgG production in non-obese individuals compared to their obese counterparts in this study, has been consistently noted in previous studies. We observed a significantly higher anti-capsular IgG titers in non-obese individuals compared to obese. The observation has been consistently noted in previous studies. David et al. reported a diminished antibody response to the hepatitis B vaccine among individuals with higher body mass index (BMI) [68, 69]. Alon et al. found significantly lower concentrations of anti-tetanus IgG in obese individuals compared to those with normal weight [69]. Similarly, Namrata et al. observed an inadequate antibody response to rabies vaccination in individuals with higher body mass index [70]. We found that non-obese individuals with diabetes exhibit higher anti-

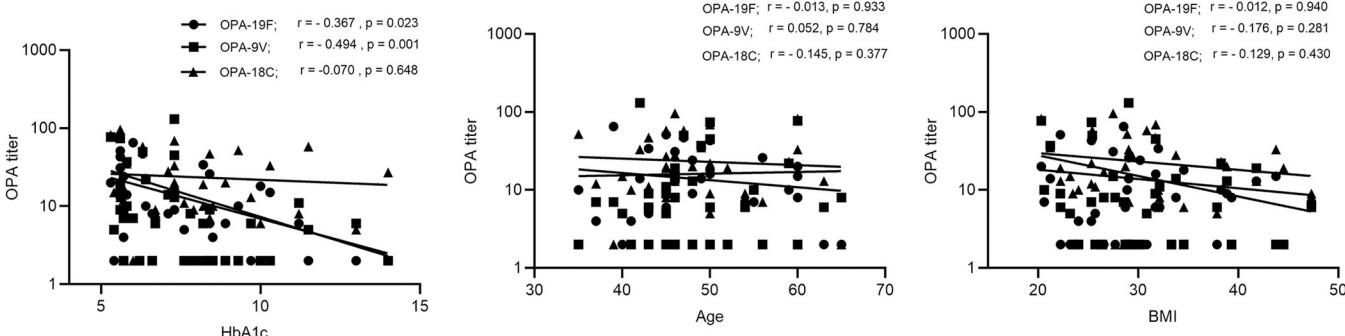

**Fig 8. Association of HbA1c, BMI, and age with opsonic titer.** Spearman (r) test was used to find a correlation between serotype-specific opsonic titers and HbA1c, BMI, and age. A significant negative correlation was observed between HbA1c and opsonic titers for 19F and 9V. A p-value < 0.05 was considered significant.

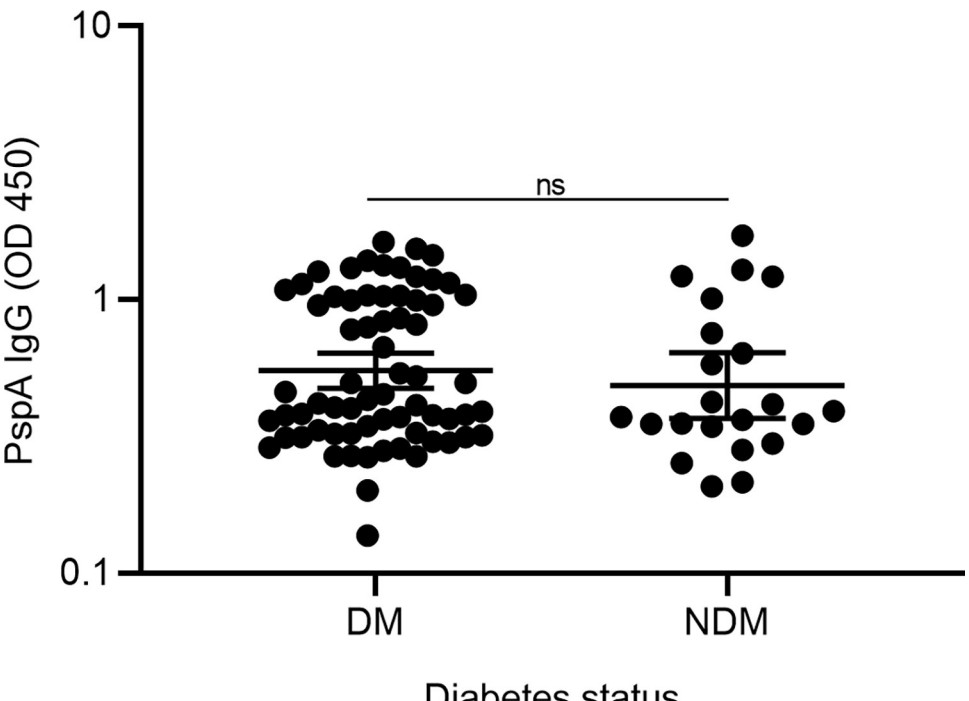

**Fig 9. Measurement of antibodies to PspA in individuals with and without type 2 diabetes.** Antibodies PspA were measured in individuals with (DM) and without type 2 diabetes (NDM) through standard ELISA. Data is represented as a geometric mean with a 95% confidence interval of the OD450 values. Mann-Whitney U test was used to compare the two groups. P-value < 0.05 was considered significant.

capsular IgG concentrations compared to their obese counterparts with diabetes. Our results align with findings from other studies that have reported alterations in antigen-specific antibody responses in obese diabetics [63, 64]. The association between obesity and reduced antibody production can be attributed to the detrimental impact of obesity on the immune system. Individuals with obesity display elevated levels of pro-inflammatory cytokines such as interleukin-6 (IL-6) and TNF-α [71–73]. The chronic elevation of these cytokines is linked to B cell defects and a dampened antigen-specific antibody response [74].

Gender and age have a considerable impact on the immune system and can affect the immune response to immunization. Several reports have highlighted the role of age and gender in response to pneumococcal vaccines as well, however the results remained inconclusive and inconsistent [75–77], in that studies reported contrasting results of unchanged, decreased, and increased antibody concentration with increasing age [52, 60, 76, 78, 79]. However, in our study we observed no significant association of antibody concentrations with increasing age. One of the reasons for the stable antibody concentrations with increasing age observed in our study may be linked to the fact that our participant's pool did not include a substantial number of individuals of advanced age. Previous studies have shown that females typically develop higher antibody responses than males to different vaccines [80]. We found elevated levels of anti-capsular IgG concentration against serotype 19F in females. Although against different serotypes, other studies have also reported increased levels of IgG in females compared to males [52, 81]. In addition, several studies have reported increased baseline antibody concentration in males compared to females [77, 79]. The contrasts in serotype-specific differences in IgG concentration between genders may arise from differential pathogen exposure, the influence of sex hormones on the immune system, and the ethnic composition of the studied population.

**Fig 10. Comparison of anti-PspA IgG (OD 450) for gender, BMI, insulin use among diabetes, and Family history of diabetes.** Data is represented as a geometric mean with a 95% confidence interval of the OD450 values. Mann-Whitney U test was used to compare two groups. P-value < 0.05 was considered significant.

A threshold of opsonic titer $\geq$ 1:8 has been shown to confer protection [28]. Although this threshold is considered widely the best functional correlate of protection [82, 83], the criteria for assessing serological correlate of protection in response to pneumococcal vaccine is not well-established in adults and particularly in those with immunocompromised conditions [84,

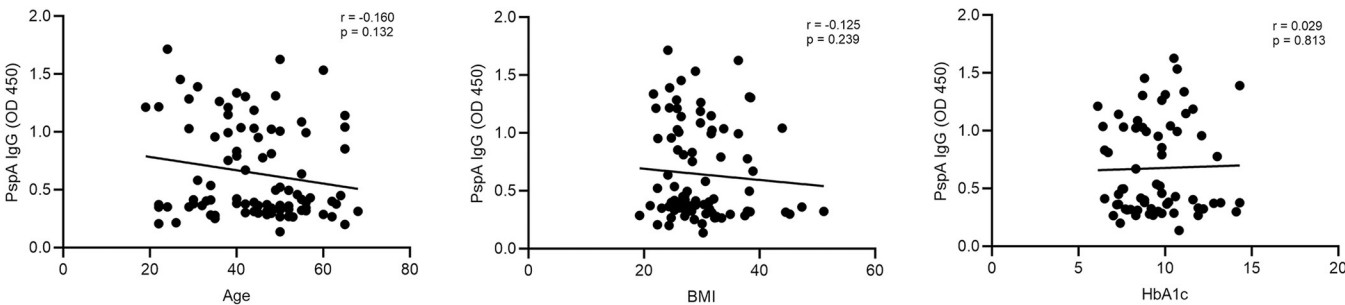

**Fig 11. Association of PspA IgG concentration with Age, BMI, and HbA1c.** Spearman correlation test (r) was used to analyze the data. A p-value <0.05 was considered significant. No statistically significant correlation was observed between IgG and any of the three variables.

85]. Studies have reported adults displaying opsonic titer of ≥ 1:8 due to natural exposure to certain pneumococcal serotypes [83]. We found that among the 3 serotypes (19F, 9V, 18C), only 18C achieved the geometric mean opsonic titer of the threshold ≥ 1:8 at baseline for the 39 samples analyzed. We observed a positive correlation with anti-capsular IgG concentration. Previous studies conducted have reported contrasting results regarding the correlation between opsonic titers and anti-capsular IgG concentration [28, 84, 86–90]. The specific cause for the discrepancy in ELISA-Opsonic titers among adults is not currently understood. Lower IgG titers and higher opsonic activities could be explained by higher antibody quality resulting in improved biological function. One of the factors that plays a crucial role in the functionality of antibodies is the glycosylation pattern as these can affect the stability, half-life, and binding characteristics of antibodies [91–93]. The reduced opsonic activity in diabetes individuals in our study aligns with previous studies, which have reported the compromised functionality of antibodies in diabetics as compared to non-diabetics for certain serotypes [12]. It is likely that hyperglycemia may glycate antibodies in addition to glycosylation resulting in inability of antibodies binding to antigen and phagocytic cell receptors [93–97]. The higher serotype-specific opsonic titers of non-obese, non-diabetes individuals in our study can be the result of the quantity and functional ability of the antibodies of these individuals. The reason for comparable opsonic titers between obese diabetes and obese non-diabetes individuals could be the comparable production of anti-capsular antibodies between obese diabetes and obese non-diabetes individuals and the role of additional influencing factors like anti-capsular IgM in the determination of opsonic titers [98–100], also suggesting that once antibodies are produced they may be equally functional in both obese and non-obese further confirming the fact that problem arises at the level of antibody production which may be the result of poor glycemic control. The observed correlation of HbA1c with opsonic titers on serotype-specific opsonic titer has been reported previously [12]. We can hypothesize that the decreased opsonic titer in the case of diabetes may be due to the hyperglycemic conditions that result in the increased glycation of antibodies, thus rendering the ability of antibodies to bind to the antigen [97, 101–103]. The other reason for the higher susceptibility of individuals with hyperglycemic conditions could be due to significant impairments in the component of the complement system [104] and also the ability of neutrophils to kill phagocytosed bacteria [105–108].

## Conclusion

In summary, using WHO reference ELISA and multiplexed opsonophagocytic killing assay, we determined anti-capsular IgG concentration and opsonic titers in non-vaccinated individuals against a few pneumococcal serotypes. The study provided us with a glimpse of how diabetes status and body mass index could shape the outcome of the production and functionality of anti-pneumococcal antibodies. This study could be the basis for evaluating immune response to pneumococcal conjugate vaccine in individuals with type 2 diabetes. The study has certain limitations, this is a single-center study with a limited sample size. We analyzed only 4 serotypes for anti-capsular IgG measurement and only 3 serotypes were analyzed for opsonic titer determination. Antibody titer to surface proteins was also measured only against Family 2 PspA and in a small number of samples. Furthermore, due to lack of pooled human serum, the study could not report titers of antibody to PspA instead the study used adjusted optical density as a measure of antibody to PspA. A larger study with reference sera will provide a better measurement of antibody titer among those with and without diabetes. Alongside, a multi-center study with a large sample size and analysis of more serotypes included in the pneumococcal conjugate vaccine will be instrumental in informing vaccine design against pneumococcal diseases for elderly and elderly with comorbid conditions.

## Supporting information

**S1 Table. Serotype-specific IgG concentration values of participants.**
(DOCX)

**S2 Table. Serotype-specific IgG concentration values of those with and without type 2 diabetes.**
(DOCX)

**S3 Table. Serotype-specific opsonic titer values of participants.**
(DOCX)

**S4 Table. Serotype-specific opsonic titer values those with and without type 2 diabetes.**
(DOCX)

**S5 Table. PspA IgG (OD 450) values of those with and without type 2 diabetes.**
(DOCX)

## Acknowledgments

We extend our sincere gratitude to Dr. Moon Nahm (University of Alabama, US) for generously providing us with pneumococcal capsular polysaccharides. Additionally, our heartfelt thanks go to Dr. Mustafa Akkoyunlu MD PhD of the United States Food and Drug Administration's Center for Biologics Evaluation and Research for supplying us with Pneumococcal Serum, US Reference Lot 007SP. We would also like to express our deep appreciation to the dedicated medical and nursing staff at the Sakine Institute of Diabetes and Endocrinology Research (SiDER), Shalamar Hospital, Lahore, Pakistan, for their invaluable assistance in the recruitment of participants.

## Author Contributions

**Conceptualization:** Izaz Ahmad, Shaper Mirza.

**Data curation:** Izaz Ahmad, Hafiz Gohar Ejaz, Rozina Arshad, Bilal Bin Younis.

**Formal analysis:** Izaz Ahmad.

**Funding acquisition:** Shaper Mirza.

**Investigation:** Izaz Ahmad, Shaper Mirza.

**Methodology:** Izaz Ahmad, Moon Nahm, Shaper Mirza.

**Project administration:** Shaper Mirza.

**Resources:** Moon Nahm.

**Software:** Robert Burton, Moon Nahm.

**Supervision:** Shaper Mirza.

**Validation:** Izaz Ahmad, Shaper Mirza.

**Visualization:** Izaz Ahmad.

**Writing – original draft:** Izaz Ahmad.

**Writing – review & editing:** Izaz Ahmad, Shaper Mirza.

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
