## [Decision Letter · Decision Letter 0]

10 May 2024

PONE-D-24-08592Naturally acquired antibodies against 4 Streptococcus pneumoniae serotypes in Pakistani adults with type 2 diabetes mellitusPLOS ONE

Dear Dr. Mirza,

Thank you for submitting your manuscript to PLOS ONE. After careful consideration, we feel that it has merit but does not fully meet PLOS ONE’s publication criteria as it currently stands. Therefore, we invite you to submit a revised version of the manuscript that addresses the points raised during the review process.

We look forward to receiving your revised manuscript.

Kind regards,

Rui Tada, Ph.D.

Academic Editor

PLOS ONE

Journal Requirements:

This study was supported by NRPU-5931-FIF, Lahore University of Managment Sciences

Reviewers' comments:

Reviewer's Responses to Questions

**Comments to the Author**

1. Is the manuscript technically sound, and do the data support the conclusions?

Reviewer #1: Yes

Reviewer #2: Yes

2. Has the statistical analysis been performed appropriately and rigorously? 

Reviewer #1: Yes

Reviewer #2: Yes

3. Have the authors made all data underlying the findings in their manuscript fully available?

Reviewer #1: Yes

Reviewer #2: No

4. Is the manuscript presented in an intelligible fashion and written in standard English?

Reviewer #1: Yes

Reviewer #2: Yes

5. Review Comments to the Author

**Reviewer #1:** The present manuscript evaluates naturally acquired antibodies against pneumococcal capsular and protein antigens in adults with or without diabetes mellitus in Pakistan. Functionality of the anti-polysaccharide antibodies was evaluated by opsonophagocytic killing assays. The data presented are very interesting and important for the field.

I have just very minor comments in an attempt to contribute to the final version.

In lines 89 and 91, the information that the serotypes chosen are responsible for around 80% of infections is repeated. Please chose to maintain the one that mentions the area (South Asia).

It would be nice to know when vaccination with PCV10 initiated in Pakistan and population coverage, since the serotypes causing such high rates of infections are included in this formulation. It is well known that PCV reduce the circulation of vaccine serotypes and induce herd immunity, protecting individuals that are not vaccinated. Therefore, one should expect that the levels of disease caused by these serotypes would be low.

Line 111 – correct antibiotic resistance strains to antibiotic resistant strains

Please indicate the ratio bacteria: cells used for the OPKA assay.

The 1.3 ug/mL threshold for the anti-PS IgG is explained in the text but I could not understand the 0.35 ug/mL threshold, shown on Figure 3. Please provide some explanation.

Line 183 – the word “observed” is repeated in the sentence.

Although the authors explain that clade 2 PspA is prevalent in the isolates from Pakistan, the evaluation of antibodies against this only clade is a limitation of the work, unless they know already the percentage of isolates expressing PspA2. The authors could comment on this limitation in the discussion.

**Reviewer #2:** This is an interesting study using WHO reference ELISA and multiplexed opsonophagocytic killing assay to determine anti-capsular IgG concentration and opsonic titers in non-vaccinated individuals in Pakistan against 4 pneumococcal serotypes. The study shows that diabetes status and body mass index influence production and functionality of anti-pneumococcal antibodies. There are some important issues in the presentation of the results that should be addressed.

Major issues

1. Table 3 (line 152) is cited before Table 2 (line 157). The text is actually a little confusing, because data are described mixing results from the two tables. I suggest describing first data in Table 2 and then data in Table 3. To do so, I suggest moving “We found no significant difference when age was stratified by diabetes status (Table 3)” in line 152 to the next paragraph. Comments on body mass index in all partipants from Table 2 should be done in the first paragraph.

2. Tables 2 and 3 use a different order of the serotypes in the columns. Using the same order makes it easier for the reader to understand results. Also in the figures, the order of serotypes is different for every figure, which is quite confusing.

3. Indication of figures in the text is also a little confusing. Indication of figures in the text should be more precise, immediately after results are described, and not two figures indicated at the end of a long paragraph. Line 168 - On comparison, no significant difference was observed in anticapsular IgG concentrations for any of the 4 serotypes between the two groups (Fig 2). When categorizing the participants into two groups based on their BMI (i) Obese; BMI > 30 kg/m2, and (ii) Non-Obese BMI: < 30 kg/m2), we found that non-obese individuals have a higher anti-capsular IgG concentration compared to obese individuals, against serotype 1, 9V and 18C in the diabetes group (Fig 3).

4. Line 175 - We analyzed the correlation of age, BMI, and HbA1c with the production of anti-capsular IgG and found that only BMI was significantly negatively correlated (r = -0.202, p = 0.014) with anti-pneumococcal serotype 1 antibodies. There were no significant correlations between any other variables with antibody concentration against any of the four serotypes – What about the analysis of correlation between BMI and anti-polysaccharide antibodies in the diabetes and non-diabetes groups?

5. Indication of figures in the text is again confusing in line 202 - Correlation analysis showed that OPA titers were positively correlated with anti-pneumococcal IgG concentrations for all the 3 serotypes (Fig 7). When OPA titers were analyzed for association with HBA1c, age, and BMI, we found that HbA1c had a statistically significant negative correlation with OPA titers for 19F and 9V (Fig 8).

6. Regarding the sentence in line 202, it seems that OPA titers were positively correlated with anti-pneumococcal IgG concentrations with statistical significance only for 19F IgG

7. Line 210 and Figure 9 – text refers to titers, whereas graphs are shown as OD 450 nm

8. Figure 10 – Since differences in anti-polysaccharide IgG between obese and non-obese individuals were observed only in diabetes patients, it would be important to stratify data on anti-PspA antibodies the same way

9. According to PLoS policy, data points behind means, medians and variance measures should be available. I could not find a link or supplemental excel sheet with the raw data of the individuals

Minor comments

10. Line 60 – correct sentence - PspA, a highly heterogenous surface protein of pneumococci, has been reported to be immunogenic, in that antibodies generated to PspA in response to colonization protect against both carriage and invasive infections.

11. Line 66 - We hypothesize that an altered baseline antibody response to asymptomatic carriage will be observed in those with diabetes as compared to those with no diabetes – Are there data in the literature on pneumococcal colonization of adults with Type 2 diabetes? In a quick search, I found one paper showing high percentage of children with Type 1 diabetes in Italy colonized with pneumococci (Hum Vaccin Immunother 2016;12(2):293-300. doi: 10.1080/21645515.2015.1072666)

12. Line 109 - To evaluate the functionality of naturally acquired anti-capsular polysaccharide we performed Multiplexed opsonophagocytic killing assay (MOPA) the using established protocol... – “the” seems to be misplaced

13. Line 122 - BSA Sigma-Aldrich – “h” is missing

6. PLOS authors have the option to publish the peer review history of their article (what does this mean?). If published, this will include your full peer review and any attached files.

Reviewer #1: No

Reviewer #2: No

---

## [Author Response · Author response to Decision Letter 0]

22 Jun 2024

Please find enclosed herewith, response to reviewers for article entitled “Naturally acquired antibodies against 4 Streptococcus pneumoniae serotypes in Pakistani adults with Type 2 diabetes mellitus” for re-evaluation and publication in PLOS ONE. The page and line number in the addressed comments are referred to in the “Revised manuscript with track changes” document. The authors are agreeable to further revision of the manuscript, if required.

We would like to thank the reviewers for sparing their valuable time and insight in reviewing our manuscript. We believe that the manuscript has been strengthened as a result. Point by point answers to the reviewers’ comments are provided as follow:

Response to Reviewer 1 Comments

Comment1: Addressed, The line mentioning South Asia was chosen. The repeated line was removed. In the revised manuscript, this issue has been addressed as follows (Page # 6, Line #89).

“These serotypes are included in 10-valent pneumococcal conjugate vaccine (PCV10) and are responsible for 80% of infection in South Asia.”

Comment 2: Thank you for this critical and constructive comment. In the revised manuscript, this issue has been addressed as follows (Page 10#, Line #147-150).

“Pakistan introduced PCV10 into its EPI in 2012. Although vaccination can result in herd immunity and lowering the disease burden, a recent publication from Pakistan shows that serotypes included in PCV10 are still among the most prevalent serotypes and are responsible for high carriage and disease burden.”

Reference: Javaid, Nida, et al. "Strain features of pneumococcal isolates in the pre-and post-PCV10 era in Pakistan." Microbial genomics 10.1 (2024): 001163

Comment 3: Addressed in the manuscript (Page #7, Line #108). 

Comment 4:Addressed in the manuscript (Page #7, Line #110).

“The ratio was 200:1 (Cells: bacteria)”

Comment 5:The 0.35 ug/mL is protective threshold of anti-PnPs IgG in children. It was used to see whether adults have the protective threshold of anti-PnPs IgG established for children. 

Comment 6: Addressed in the manuscript (Page #15, Line #201).

Comment 7: Addressed in conclusion of the study. Limitation added. (Page #24, Line #322).

Response to Reviewer 2 Comments

Comment 1. Addressed by moving the relevant text to correct positions in the paragraphs. (Changes can be tracked on Page # 11, Line #155- 163). 

Comment 2: Addressed the issue by changing the order of serotypes in Tab 2 on Page #12-13 and Figures

Comment 3: Addressed by moving Figure 2 and Fig 3 indications into relevant positions in the test as suggested.

(Page # 11, line159, 164).

Comment 4: Thanks to the reviewer for this valuable comment. The comment is addressed, kindly refer to (Page #15, Line #193 - 195). The following analysis was carried out and added to the paper. 

“When analyzed correlation between BMI and anti-PnPs IgG in diabetes and non-diabetes groups individually, we found significant negative correlation only between anti-PnPs1 IgG and BMI in the diabetes group. No significant correlation was found between anti-PnPs IgG and BMI in the non-diabetes group. The statement has been added to the paper.” 

Comment 5: Addressed by moving the “figure indications to correct position as suggested 

(Line# 226 & 230 Page # 18).

Comment 6: Yes, that is correct. We’ve changed the statement and is now written more clearly regarding the statistically significant association for only serotype 19F. The statement now goes as follows in the revised version (Page #18, Line# 225 -226)

“Although correlation analysis showed that OPA titers were positively correlated with anti-pneumococcal IgG concentrations for all the 3 serotypes, the association was found to be statistically significant only for 19F (Fig 7).”

Comment 7: The issue is addressed. Provided explanation in lines # 233-236, page # 18. Removed the word titer in the text. (line # 237 …, Page # 18-19). Also, put as a limitation in the conclusion section (line # 323-325, Page# 24)

Reference: Melin, Merit M., et al. "Development of antibodies to PspA families 1 and 2 in children after exposure to Streptococcus pneumoniae." Clinical and Vaccine Immunology 15.10 (2008): 1529-1535.

Comment 8: The analysis was performed and added to the revised manuscript (Page #19, Line# 250 -252) as follows 

“On stratification of diabetes status by BMI, we found that obese diabetes individuals (GM: 0.497, 95% CI: 0.393 – 0.629) have comparable IgG with those of obese non-diabetes (GM: 0.334, 95% CI: 0.094 – 1.178). Also, non-obese diabetes (GM: 0.612, 95% CI: 0.503 – 0.743) were found to have comparable IgG with those of non-obese non-diabetes (GM: 0.514, 95% CI: 0.380 – 0.695).”

Comment 9: The data points are included as an additional document with the resubmitted version.

Comment 10: Addressed and revised the sentence (page# 4 line# 59-60). The statement now goes as follows

“PspA, a highly heterogeneous surface protein of pneumococci, has been reported to be immunogenic, and antibodies generated to PspA in response to colonization protect against carriage and invasive infections [38-41].”

Comment 11: Although literature has reported the higher susceptibility, hospitalization, and mortality in diabetes patients due to pneumococcal infections, there is no exclusive study done on pneumococcal colonization in adults with type 2 diabetes. 

Comment 12: Addressed the issue (page #7, line # 107)

Comment 13: Addressed the issue (page # 8, line # 120)

---

## [Editor Report · Decision Letter 1]

26 Jun 2024

Naturally acquired antibodies against 4 Streptococcus pneumoniae serotypes in Pakistani adults with type 2 diabetes mellitus

PONE-D-24-08592R1

Dear Dr. Mirza,

We’re pleased to inform you that your manuscript has been judged scientifically suitable for publication and will be formally accepted for publication once it meets all outstanding technical requirements.

Kind regards,

Rui Tada, Ph.D.

Academic Editor

PLOS ONE

---

## [Editor Report · Acceptance letter]

1 Jul 2024

PONE-D-24-08592R1 

PLOS ONE

Dear Dr. Mirza, 

I'm pleased to inform you that your manuscript has been deemed suitable for publication in PLOS ONE. Congratulations! Your manuscript is now being handed over to our production team.

Kind regards, 

on behalf of

Dr. Rui Tada 

Academic Editor

PLOS ONE